# PromptIR: Prompting for All-in-One Blind Image Restoration

**Vaishnav Potlapalli**[*], **Syed Waqas Zamir**[†]**, Salman Khan**[*]**, Fahad Shahbaz Khan**[*‡]
[*]Mohamed bin Zayed University of AI, [†]Core42, [‡]Linköping University
`firstname.lastname@mbzuai.ac.ae`

## Abstract

Image restoration involves recovering a high-quality clean image from its degraded version. Deep learning-based methods have significantly improved image restoration performance, however, they have limited generalization ability to different degradation types and levels. This restricts their real-world application since it requires training individual models for each specific degradation and knowing the input degradation type to apply the relevant model. We present a prompt-based learning approach, PromptIR, for All-In-One image restoration that can effectively restore images from various types and levels of degradation. In particular, our method uses prompts to encode degradation-specific information, which is then used to dynamically guide the restoration network. This allows our method to generalize to different degradation types and levels, while still achieving state-of-the-art results on image denoising, deraining, and dehazing. Overall, PromptIR offers a generic and efficient plugin module with few lightweight prompts that can be used to restore images of various types and levels of degradation with no prior information on the corruptions present in the image. Our code and pre-trained models are available here: `https://github.com/va1shn9v/PromptIR`.

## 1 Introduction

During image acquisition, degradations (such as noise, blur, haze, rain, etc.) are often introduced either due to the physical limitations of cameras or unsuitable ambient conditions. Image restoration refers to the process of recovering a high-quality clean image from its degraded version. It is a highly challenging problem due to its ill-posed nature as there exists many feasible solutions, both natural and unnatural. Recently, deep learning based restoration approaches [47, 12, 69, 45, 76, 55, 43, 74] have emerged as more effective choice in comparison to conventional methods [19, 36, 13, 53, 27, 42, 28].

Deep neural network-based methods broadly differ in their approach to addressing the image restoration problem. Some works incorporate explicit task-specific knowledge in the network to deal with the corresponding restoration task, such as denoising [45, 76], deblurring [43, 74], and dehazing [47, 12, 36]. However, these methods lack generalization beyond the specific degradation type and level. On the other hand, some works [56, 68, 59, 71, 70, 8] focus on developing a robust architecture design and learn image priors from data implicitly. These methods train separate copies of the same network for different degradation types, degradation levels, and in more extreme cases on different datasets. However, replicating the same restoration model for different degradation types, levels, and data distributions is a compute-intensive and tedious process, and oftentimes impractical for resource-constrained platforms like mobile and edge devices. Furthermore, to select an appropriate restoration model during testing, these approaches require prior knowledge regarding the degradation present in the input image.

37th Conference on Neural Information Processing Systems (NeurIPS 2023).

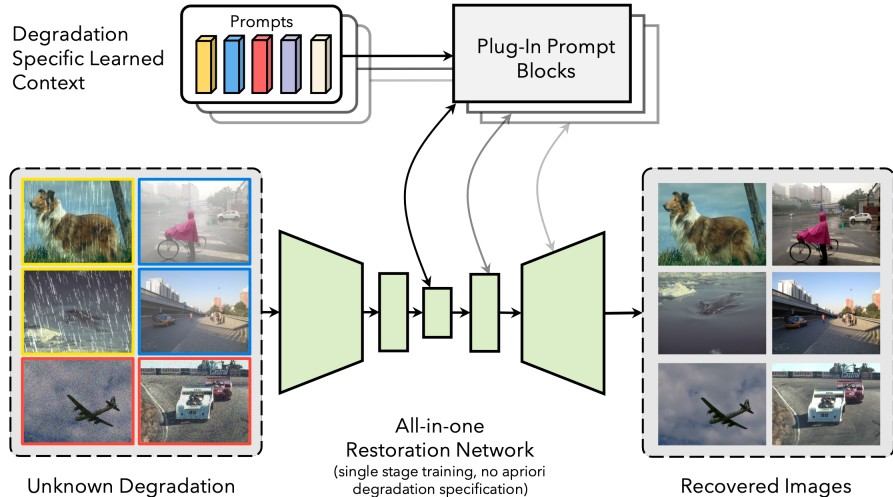

Figure 1: This figure illustrates our PromptIR approach. We propose a plug-and-play prompt module that implicitly predicts degradation-conditioned prompts to guide the restoration process of an input image with unknown degradation. The guidance from prompts is injected into the network at multiple decoding stages with few-learnable parameters. This allows learning an all-in-one unified model that can perform well across several image restoration tasks (e.g., draining, dehazing, and denoising).

Therefore, there is a pressing need to develop an *all-in-one* method that can effectively restore images from various types and levels of degradation.

One recent method, AirNet [29], addresses the all-in-one restoration task by employing the contrastive learning paradigm. This involves training an extra encoder to differentiate various types of image degradations. Although AirNet [29] yields state-of-the-art results, it struggles to model fully disentangled representations of different corruption types. Furthermore, the usage of an additional encoder for contrastive learning leads to a higher training burden due to the two-stage training approach.

To overcome these challenges, in this paper, we present a prompt-learning-based approach to perform all-in-one image restoration (see Fig. 1). Our method utilizes prompts, which are a set of tunable parameters that encode crucial discriminative information about various types of image degradation (as shown in Fig. 2). By interacting prompts with the feature representations of the main restoration network, we dynamically enhance the representations with degradation-specific knowledge. This adaptation enables the network to effectively restore images by dynamically adjusting its behavior. The main highlights of our work include,

- We present a prompting-based all-in-one *blind* restoration framework PromptIR that relies solely on the input image to recover a clean image, without requiring any prior knowledge of the degradation present in the image.

- Our prompt block is a plug-in module that can be easily integrated into any existing restoration network. It consists of a prompt generation module (PGM) and a prompt interaction module (PIM). The goal of the prompt block is to generate input-conditioned prompts (via PGM) that are equipped with useful contextual information to guide the restoration network (with PIM) to effectively remove the corruption from the input image.

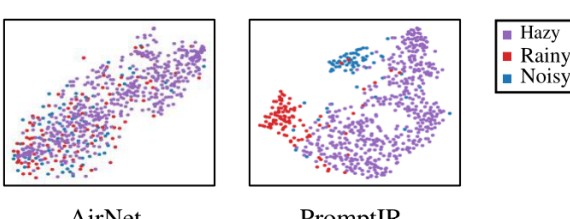

AirNet          PromptIR

Figure 2: The figure shows tSNE plots of the degradation embeddings used in PromptIR (ours) and the state-of-the-art AirNet [29]. Distinct colors denote different degradation types. In our case, the embeddings for each task are better clustered, showing the effectiveness of prompt tokens to learn discriminative degradation context that helps in restoration.

- Our comprehensive experiments demonstrate the dynamic adaptation behavior of PromptIR by achieving state-of-the-art performance on various image restoration tasks, including image denoising, deraining, and dehazing using only a *unified* PromptIR model.

## 2 Related Works

**Multi-degradation Image Restoration:** While single degradation image restoration methods [68, 47, 12, 69, 45, 76, 55, 43, 74, 51] have received significant interest, multi-degradation image restoration is relatively under-explored in the literature. A body of work focuses on images corrupted due to multiple weather conditions e.g., snow, fog, and rain [37, 57, 32]. However, they train specific encoder or decoder parallel pathways for each weather degradation which requires knowing specific degradation type and is less scalable. Chen *et al.* [6] build a unified model for multiple restoration tasks, like super-resolution, denoising, and deraining, however, the model needs prior information about the corruption present in the input image as it uses a multi-head-tail architecture. In blind image restoration, we have no prior information on the degradation present in the image. This kind of problem setting has been tackled in the context of image super-resolution [73, 39, 10]. Li *et al.* [32] introduce a unified model for denoising, draining, and dehazing, which uses an image encoder trained through contrastive learning to model good representations of the degradation, which are later used to predict the deformable convolution offsets in another network to perform the restoration. This method requires two-stage training and the effectiveness of contrastive learning hinges on accurately choosing the positive-negative pairs and the amount of data available. In comparison, our work is focused on developing a single-stage training pipeline for unified all-in-one image restoration that is conceptually simpler and works as a drop-in module for multiple degradations.

**Transformer-based restoration:** Transformer [58] architectures have found great success across various computer vision tasks [25] such as image recognition [15, 54, 67], object detection [5, 80, 38] and semantic segmentation [64, 59, 78]. Owing to their strong feature representation capability, they are extended to image restoration tasks [7, 59, 56, 9]. However, naive self-attention has quadratic complexity w.r.t. the image size and this poses a challenge for image restoration tasks where inputs are typically high-resolution. To address this, some works have proposed efficient transformer architectures [33, 68, 35] to reduce the computational costs. Specifically, SwinIR [35] uses windowed self-attention blocks along with convolutional layers to improve the efficiency of the model. Restormer [68] uses multi-depth convolution head attention to reduce the number of operations. In this work, we apply our PromptIR to Restormer owing to its efficient design and high performance, however, our prompt block is generic and can work with other architectures.

**Prompt learning:** In natural language processing, prompting-based methods are means to provide in-context information to models to finetune them on a target task [3]. However, instead of using specific manual instruction sets as prompts, learnable prompts enable better parameter-efficient adaptation of models [79]. Prompt learning techniques can effectively model task-specific context hence they have been used for finetuning to vision tasks [23, 34, 26] and incremental learning [62, 49, 61]. Prompt learning-based techniques have also been applied in the case of multitask learning [20, 60], where choosing the right prompt for each task remains critical. All these approaches target high-level vision problems, however, our goal here is to develop a generic model for low-level vision that can dynamically restore inputs based on their interaction with the prompts. The prompts act as an adaptive lightweight module to encode degradation context across multiple scales in the restoration network.

## 3 Method

In "All-in-one" image restoration, we aim to learn a single model $M$ to restore an image $I$ from a degraded image $\tilde{I}$, that has been degraded using a degradation $D$, while having no prior information about $D$. While the model is initially "blind" to the nature of degradation, its performance in recovering a clean image can be enhanced by providing implicit contextual information about the type of degradation. In this paper, we present prompt learning-based image restoration framework **PromptIR**, shown in Fig. 3. Prompting is an efficient[23] and suitable[20] method for supplementing the model with relevant knowledge of the degradation type while recovering the clean image. The key element of PromptIR is the prompt blocks that first generate learnable prompt parameters, and then use these prompts to guide the model during the restoration process. Next, we describe the overall pipeline of our PromptIR framework and its components in detail.

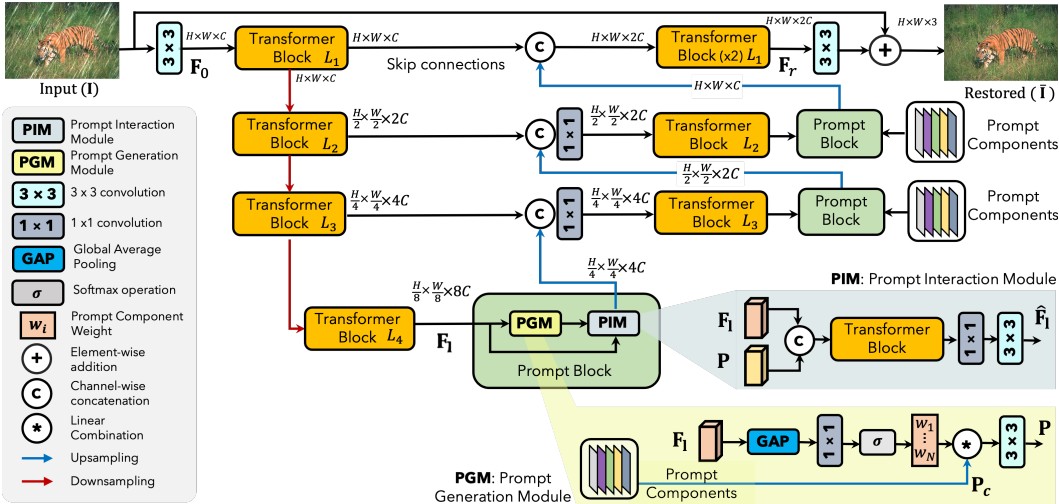

Figure 3: Overview of the PromptIR approach. We use a UNet-style network [68] with transformer blocks in the encoding and decoding stages. The primary component of the framework, i.e., the prompt block consists of two modules, the Prompt Generation Module (PGM) and the Prompt Interaction Module (PIM). The prompt generation module generates the input-conditioned prompt $\mathbf{P}$, using the input features $\mathbf{F}_l$ and the Prompt Components. The prompt interaction module then dynamically adapts the input features using the generated prompt through the transformer block. The prompts interact with decoder features at multiple levels to enrich the degradation-specific context.

**Overall pipeline.** From a given degraded input image $\mathbf{I} \in \mathbb{R}^{H \times W \times 3}$, PromptIR first extracts low-level features $\mathbf{F_0} \in \mathbb{R}^{H \times W \times C}$ by applying a convolution operation; where $H \times W$ is the spatial resolution and $C$ denotes the channels. Next, the feature embeddings $\mathbf{F_0}$ undergo a 4-level hierarchical encoder-decoder, transforming into deep features $\mathbf{F_r} \in \mathbb{R}^{H \times W \times 2C}$. Each level of the encoder-decoder employs several Transformer blocks, with the number of blocks gradually increasing from the top level to the bottom level to maintain computational efficiency. Starting from the high-resolution input, the goal of the encoder is to progressively reduce the spatial resolution while increasing channel capacity, thereby yielding low-resolution latent representation $\mathbf{F}_l \in \mathbb{R}^{\frac{H}{8} \times \frac{W}{8} \times 8C}$. From the low-resolution latent features $\mathbf{F}_l$, the aim of the decoder is to gradually recover the high-resolution clean output. In order to assist the decoding process, we incorporate prompt blocks in our PromptIR framework. Prompt blocks are adapter modules that sequentially connect every two levels of the decoder. At each decoder level, the prompt block implicitly enriches the input features with information about the degradation type for a guided recovery. Next, we describe the proposed prompt block and its core building modules in detail.

### 3.1 Prompt Block

In NLP [3, 48, 21, 34] and vision tasks [23, 26, 18, 50], prompting-based techniques have been explored for parameter-efficient finetuning of large frozen models trained on a source task $\mathcal{S}$ onto a target task $\mathcal{T}$. The effective performance of prompting-based techniques is attributed to their ability to efficiently encode task-specific contextual information in prompt components. In the proposed PromptIR, prompt components are learnable parameters, that interact with the input features in order to enrich them with degradation type. Given $N$ prompt-components $\mathbf{P_c} \in \mathbb{R}^{N \times \hat{H} \times \hat{W} \times \hat{C}}$ and input features $\mathbf{F_l} \in \mathbb{R}^{\hat{H} \times \hat{W} \times \hat{C}}$, the overall process of prompt block is defined as:

$$\hat{\mathbf{F}}_\mathbf{l} = \texttt{PIM}(\texttt{PGM}(\mathbf{P_c}, \mathbf{F_l}), \mathbf{F_l}) \tag{1}$$

The prompt block consists of two key components: a prompt generation module (PGM) and a prompt-interaction module (PIM), each of which we describe next.

### 3.1.1 Prompt Generation Module (PGM)

Prompt components $\mathbf{P_c}$ form a set of learnable parameters that interact with the incoming features to embed degradation information. One straightforward method for features-prompt interaction is to directly use the learned prompts to calibrate the features. However, such a static approach may yield suboptimal results, as it is agnostic to the input content. Therefore, we present PGM that dynamically predicts attention-based weights from the input features and apply them to prompt components to yield input-conditioned prompts $\mathbf{P}$. Furthermore, PGM creates a shared space to facilitate correlated knowledge sharing among prompt components.

To generate prompt-weights from the input features $\mathbf{F_l}$, PGM first applies global average pooling (GAP) across spatial dimension to generate feature vector $\mathbf{v} \in \mathbb{R}^{\hat{C}}$. Next, we pass $\mathbf{v}$ through a channel-downscaling convolution layer to obtain a compact feature vector, followed by the softmax operation, thus yielding prompt-weights $w \in \mathbb{R}^N$. Finally, we use these weights to make adjustments in prompt components, followed by a $3 \times 3$ convolution layer. Overall, the PGM process is summarized as:

$$\mathbf{P} = \mathtt{Conv}_{3 \times 3}\left( \sum_{c=1}^{N} w_c \mathbf{P}_c \right), \qquad w = \mathtt{Softmax}(\mathtt{Conv}_{1 \times 1}(\mathtt{GAP}(\mathbf{F_l}))) \qquad (2)$$

Since at inference time, it is necessary for the restoration network to be able to handle images of different resolutions, we cannot use the prompt components $\mathbf{P_c}$ with a fixed size. Therefore, we apply the bilinear upsampling operation to upscale the prompt components to the same size as the incoming input features.

### 3.1.2 Prompt Interaction Module (PIM)

The primary goal of PIM is to enable interaction between the input features $\mathbf{F_l}$ and prompts $\mathbf{P}$ for a guided restoration.

In PIM, we concatenate the generated prompts with the input features along the channel dimension. Next, we pass the concatenated representations through a Transformer block that exploits degradation information encoded in the prompts and transforms the input features.

The main contribution of this paper is the prompt block, which is a plug-in module, and architecture agnostic. Therefore, in the proposed PromptIR framework, we use an existing Transformer block [68], instead of developing a new one. The Transformer block is composed of two sequentially connected sub-modules: Multi-Dconv head transposed attention (MDTA), and Gated-Dconv feedforward network (GDFN). MDTA applies self-attention operation across channels rather than the spatial dimension and has linear complexity. The goal of GDFN is to transform features in a controlled manner, i.e., suppressing the less informative features and allowing only useful ones to propagate through the network. The overall process of PIM is:

$$\hat{\mathbf{F}}_l = \mathtt{Conv}_{3 \times 3}(\mathtt{GDFN}(\mathtt{MDTA}[\mathbf{F_l}; \mathbf{P}])) \qquad (3)$$

where [ ; ] is concatenation operation. MDTA is formulated as $\mathbf{Y} = W_p \mathbf{V} \cdot \mathtt{Softmax}\left(\mathbf{K} \cdot \mathbf{Q}/\alpha\right) + \mathbf{X}$. Where $\mathbf{X}$ and $\mathbf{Y}$ are the input and output features. $\mathbf{Q}$, $\mathbf{K}$ and $\mathbf{V}$ respectively represent query, key, and value projections that are obtained by applying $1 \times 1$ point-wise convolutions followed by $3 \times 3$ depth-wise convolutions on the layer normalized input feature maps. $W_p$ is the point-wise convolution, $\alpha$ denotes a learnable scaling parameter, and $(\cdot)$ represents dot-product interaction. The process of GDFN is defined as $\mathbf{Z} = W_p^0 \left( \phi(W_d^1 W_p^1(\mathtt{LN}(\mathbf{Y}))) \odot W_d^2 W_p^2(\mathtt{LN}(\mathbf{Y})) \right) + \mathbf{Y}$. Where, $W_d^{(\cdot)}$ is the $3 \times 3$ depth-wise convolution, $\odot$ denotes element-wise multiplication, $\phi$ is the GELU non-linearity, and LN is the layer normalization [2]. The block diagram and additional details on the Transformer block are provided in the appendix.

## 4 Experiments

To demonstrate the effectiveness of the proposed PromptIR, we perform the evaluation on three representative image restoration tasks: image dehazing, image deraining, and image denoising. Following [29], we conduct experiments under two different experimental settings: **(a)** All-in-One, and **(b)** Single-task.

Table 1: Comparisons under All-in-one restoration setting: single model trained on a combined set of images originating from different degradation types. When averaged across different tasks, our PromptIR provides a significant gain of 0.86 dB over the previous all-in-one method AirNet [29].

| Method | Dehazing on SOTS [31] | Deraining on Rain100L [16] | Denoising on BSD68 dataset [41]) | | | Average |
| | | | $\sigma = 15$ | $\sigma = 25$ | $\sigma = 50$ | |
| --- | --- | --- | --- | --- | --- | --- |
| BRDNet [52] | 23.23/0.895 | 27.42/0.895 | 32.26/0.898 | 29.76/0.836 | 26.34/0.836 | 27.80/0.843 |
| LPNet [17] | 20.84/0.828 | 24.88/0.784 | 26.47/0.7782 | 24.77/0.748 | 21.26/0.552 | 23.64/0.738 |
| FDGAN [14] | 24.71/0.924 | 29.89/0.933 | 30.25/0.910 | 28.81/0.868 | 26.43/0.776 | 28.02/0.883 |
| MPRNet [71] | 25.28/0.954 | 33.57/0.954 | 33.54/0.927 | 30.89/0.880 | 27.56/0.779 | 30.17/0.899 |
| DL[16] | 26.92/0.391 | 32.62/0.931 | 33.05/0.914 | 30.41/0.861 | 26.90/0.740 | 29.98/0.875 |
| AirNet [29] | 27.94/0.962 | 34.90/0.967 | 33.92/0.933 | 31.26/0.888 | 28.00/0.797 | 31.20/0.910 |
| PromptIR (Ours) | **30.58/0.974** | **36.37/0.972** | **33.98/0.933** | **31.31/0.888** | **28.06/0.799** | **32.06/0.913** |

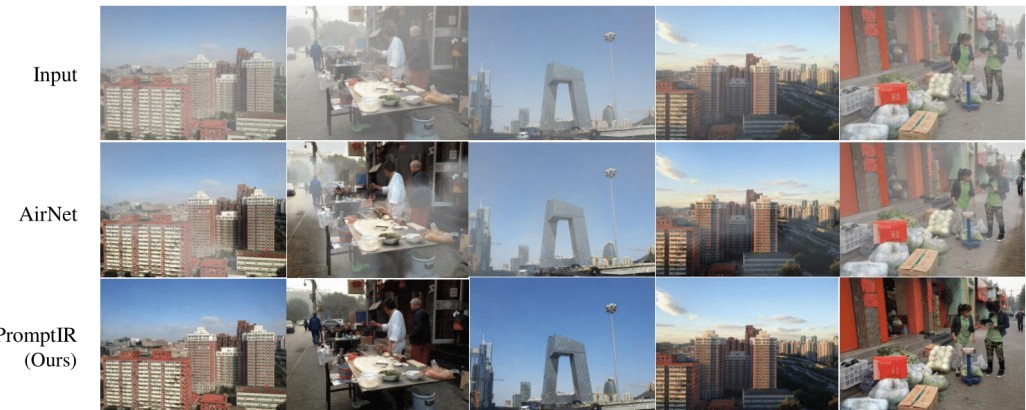

Figure 4: **Dehazing comparisons** for all-in-one methods on images from the SOTS dataset [31]. The image quality of the results produced by our PromptIR is visually better than the previous state-of-the-art approach AirNet[29].

In the All-in-One setting, we train a unified model that can recover images across all three degradation types. Whereas, for the Single-task setting, we train separate models for different restoration tasks. The image quality scores for the best and second-best methods are **highlighted** and underlined in the tables.

**Implementation Details.** Our PromptIR framework is end-to-end trainable and requires no pre-training of any individual component. The architecture of our PromptIR consists of a 4-level encoder-decoder, with varying numbers of Transformer blocks at each level, specifically [4, 6, 6, 8] from level-1 to level-4.

We employ one prompt block between every two consecutive decoder levels, totaling 3 prompt blocks in the overall PromptIR network. The total number of prompt components are 5. The model is trained with a batch size of 32 in the all-in-one setting, and with a batch of 8 in the single-task setting. The network is optimized with an $L_1$ loss, and we use Adam optimizer ($\beta_1 = 0.9$, $\beta_2 = 0.999$) with learning rate $2e - 4$ for 200 epochs. During training, we utilize cropped patches of size 128 x 128 as input, and to augment the training data, random horizontal and vertical flips are applied to the input images.

**Datasets.** We prepare datasets for different restoration tasks, following closely the prior work [29]. For image denoising in the single-task setting, we use a combined set of BSD400 [1] and WED [40] datasets for training. The BSD400 dataset contains 400 training images and the WED dataset has 4,744 images. From clean images of these datasets, we generate the noisy images by adding Gaussian noise with different noise levels $\sigma \in \{15, 25, 50\}$. Testing is performed on BSD68 [41] and Urban100 [22] datasets. For single-task image deraining, we use the Rain100L [65] dataset, which consists of 200 clean-rainy image pairs for training, and 100 pairs for testing. Finally, for image dehazing in the single-task setting, we utilize SOTS [31] dataset that contains 72,135 training images

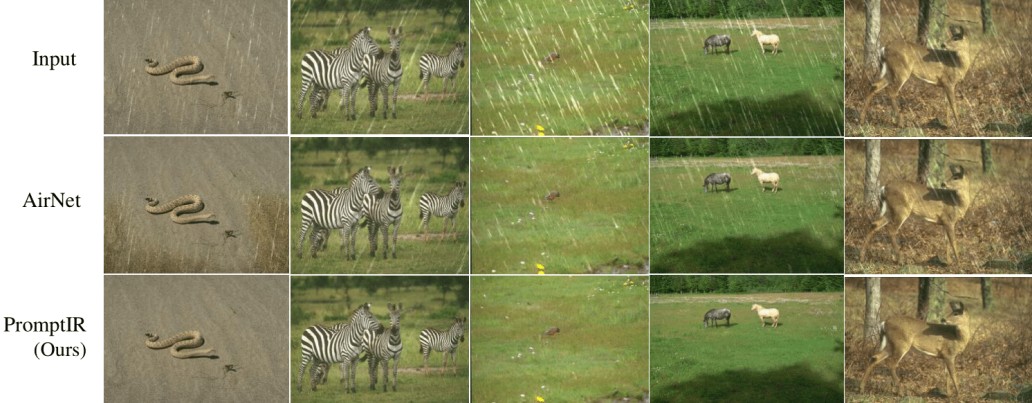

Figure 5: **Image deraining comparisons** for all-in-one methods on images from the Rain100L dataset [16]. Our method effectively removes rain streaks to generate rain-free images.

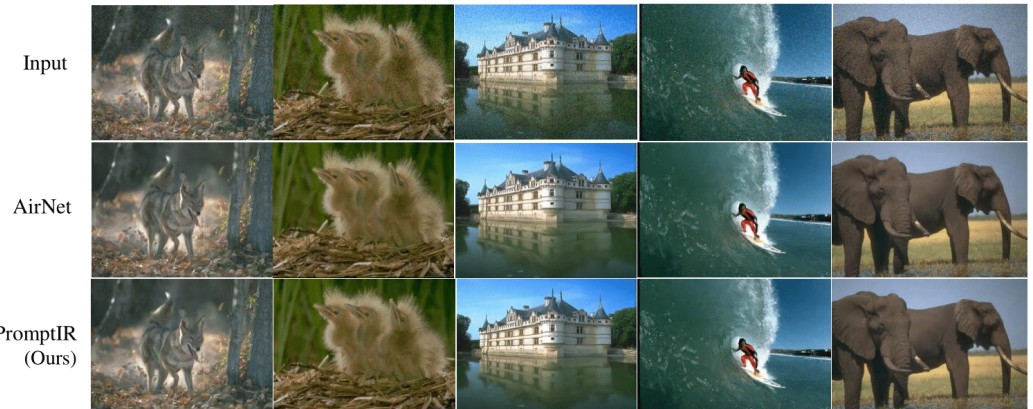

Figure 6: **Denoising results** for all-in-one methods.

and 500 testing images. Finally, to train a unified model under the all-in-one setting, we combine all 4 aforementioned datasets and train a single model that is later evaluated on multiple tasks.

### 4.1 Multiple Degradation All-in-One Results

We compare the proposed PromptIR with several general image restoration approaches [52, 17, 14, 71] as well as with specialized all-in-one methods [16, 29]. Results are reported in Table 1. When averaged across different restoration tasks, our algorithm yields 0.86 dB performance gain over the previous best method AirNet [29], and 1.89 dB over the second best approach DL [71]. Specifically, the proposed PromptIR significantly advances state-of-the-art by providing 2.64 dB PSNR improvement on the image dehazing task. The visual examples provided in Fig. 4 show that PromptIR effectively removes haze from the input images, and generates cleaner results than AirNet [29]. Similarly on the image deraining task as shown in Table 1, the proposed PromptIR achieves a substantial gain of 3.73 dB compared to DL [16] and 1.47 dB over AirNet [29]. Visual comparisons in Fig. 5 show that PromptIR is capable of removing rain streaks of various orientations and generates visually pleasant rain-free images. Finally, on the denoising task, our method provides 1.16 db boost over the DL algorithm [16] for a high noise level of $\sigma$=50. Qualitative examples are presented in Fig. 6, where our method reproduces noise-free images with better structural fidelity than the AirNet algorithm [29].

Table 2: Dehazing results in the single-task setting on the SOTS benchmark dataset [31]. Our PromptIR achieves a significant boost of 8.13 dB over AirNet [29].

| Method | DehazeNet[4] | MSCNN[46] | AODNet[30] | EPDN[44] | FDGAN[14] | AirNet[29] | Restormer[68] | PromptIR (Ours) |
|---|---|---|---|---|---|---|---|---|
| PSNR | 22.46 | 22.06 | 20.29 | 22.57 | 23.15 | 23.18 | 30.87 | **31.31** |
| SSIM | 0.851 | 0.908 | 0.877 | 0.863 | 0.921 | 0.900 | 0.969 | **0.973** |

Table 3: Deraining results in the single-task setting on Rain100L [16]. Compared to the AirNet [29] algorithm, the proposed method yields 2.13 dB PSNR improvement.

| Method | DIDMDN[72] | UMR[66] | SIRR[63] | MSPFN[24] | LPNet[17] | AirNet[29] | Restormer[68] | PromptIR(Ours) |
|---|---|---|---|---|---|---|---|---|
| PSNR | 23.79 | 32.39 | 32.37 | 33.50 | 33.61 | 34.90 | 36.74 | **37.04** |
| SSIM | 0.773 | 0.921 | 0.926 | 0.948 | 0.958 | 0.977 | 0.978 | **0.979** |

Table 4: Denoising comparisons in the single-task setting on BSD68 [41] and Urban100 [22] datasets. For the challenging noise level of $\sigma = 50$ on Urban100 [22], our PromptIR obtains 0.51 dB gain compared to AirNet [29].

| Method | BSD68 [41] | | | Urban100 [22] | | |
|---|---|---|---|---|---|---|
| | $\sigma = 15$ | $\sigma = 25$ | $\sigma = 50$ | $\sigma = 15$ | $\sigma = 25$ | $\sigma = 50$ |
| CBM3D [11] | 33.50/0.922 | 30.69/0.868 | 27.36/0.763 | 33.93/0.941 | 31.36/0.909 | 27.93/0.840 |
| DnCNN [75] | 33.89/0.930 | 31.23/0.883 | 27.92/0.789 | 32.98/0.931 | 30.81/0.902 | 27.59/0.833 |
| IRCNN [76] | 33.87/0.929 | 31.18/0.882 | 27.88/0.790 | 27.59/0.833 | 31.20/0.909 | 27.70/0.840 |
| FFDNet [77] | 33.87/0.929 | 31.21/0.882 | 27.96/0.789 | 33.83/0.942 | 31.40/0.912 | 28.05/0.848 |
| BRDNet [52] | 34.10/0.929 | 31.43/0.885 | 28.16/0.794 | 34.42/0.946 | 31.99/0.919 | 28.56/0.858 |
| AirNet [29] | 34.14/0.936 | 31.48/0.893 | 28.23/0.806 | 34.40/0.949 | 32.10/0.924 | 28.88/0.871 |
| PromptIR(Ours) | **34.34/0.938** | **31.71/0.897** | **28.49/0.813** | **34.77**/0.952 | **32.49/0.929** | **29.39/0.881** |

## 4.2 Single Degradation One-by-One Results

In this section, we evaluate the performance of our PromptIR under the single-task setting, i.e., a separate model is trained for different restoration tasks. This is to show that content-adaptive prompting via prompt block is also useful for single-task networks. Table 2 presents dehazing results. It shows that our PromptIR achieves 8.13 dB improvement over AirNet [29], and 0.44 dB gain over the baseline method Restormer [68]. Similar trends can be observed for deraining and denoising tasks. For instance, when compared to the AirNet [29], our method yields performance gains of 2.13 dB on the deraining task (Table 3) and 0.51 dB on denoising task for noise level $\sigma$=50 on Urban100 dataset [22] (see Table 4).

## 4.3 Ablations Studies

We perform several ablation experiments to demonstrate that our contributions in PromptIR framework provides quality improvements.

**Impact of PGM.** We carry out this ablation experiment on Rain100L [65] for deraining task. Table 5 shows that the prompt block in our PromptIR network brings performance gains of 0.3 dB over the baseline [68]. Further, it demonstrates that generating dynamic prompts conditioned on input content via PGM provides a favorable gain of 0.19 dB over the fixed prompt components.

**Position of prompt blocks.** In the hierarchical architecture of our PromptIR, we analyze where to place prompt blocks on the decoder side. Table 6 shows that using only one prompt block in the latent space degrades the network's performance. Whereas, incorporating prompt blocks between every consecutive level of the decoder performs the best.

**Generalization to unseen degradation level.** We take the model that is trained only on the noise levels $\sigma \in \{15, 25, 50\}$ and test its performance on the unseen noise level of $\sigma = 100$.

Table 7 shows that our PromptIR demonstrates significantly superior generalization capabilities compared to AirNet [29], yielding ~7 dB PSNR difference.

Table 5: Impact of PGM. Results are reported on Rain100L [65] dataset.

| Method | PSNR |
|---|---|
| Baseline [68] | 36.74 |
| Fixed-Prompt | 36.85 |
| Dynamic-Prompt | 37.04 |

Table 6: Prompt blocks position. Results are reported on Rain100L [65] dataset.

| Model | PSNR |
|---|---|
| level 4 (latent) | 36.76 |
| levels 4+3 | 36.84 |
| levels 4+3+2 | 37.04 |

Table 7: Denoising comparisons on unseen noise level of $\sigma = 100$.

| | BSD68 [41] | Urban100 [22] |
|---|---|---|
| Method | PSNR | PSNR |
| Airnet [29] | 13.64 | 13.72 |
| PromptIR (Ours) | **21.03** | **20.50** |

Table 8: Evaluation on Spatially Variant Degradation on BSD68 [41] test set.

| Model | PSNR |
|---|---|
| Airnet [29] | 31.42 |
| PromptIR (Ours) | **31.65** |

Table 9: Performance of the proposed PromptIR framework, when trained on different combinations of degradation types (tasks) i.e., removal of noise, rain and haze.

| Degradation | | | Denoising on BSD68 dataset [41] | | | Deraining on Rain100L [16] | Dehazing on SOTS [31] |
|---|---|---|---|---|---|---|---|
| Noise | Rain | Haze | $\sigma = 15$ | $\sigma = 25$ | $\sigma = 50$ | | |
| ✓ | ✗ | ✗ | 34.34/0.938 | 31.71/0.898 | 28.49/0.813 | - | - |
| ✗ | ✓ | ✗ | - | - | - | 37.03/0.9786 | - |
| ✗ | ✗ | ✓ | - | - | - | - | 31.31/0.929 |
| ✓ | ✓ | ✗ | 34.26/0.937 | 31.61/0.895 | 28.37/0.810 | 39.32/0.986 | - |
| ✓ | ✗ | ✓ | 33.69/0.928 | 31.03/0.880 | 27.74/0.777 | - | 30.09/0.975 |
| ✗ | ✓ | ✓ | - | - | - | 35.12/0.969 | 30.36/0.973 |
| ✓ | ✓ | ✓ | 33.98/0.933 | 31.31/0.888 | 28.06/0.799 | 36.37/0.972 | 30.58/0.974 |

**Performance on spatially variant degradation.** Here we evaluate PromptIR performance on images that are corrupted with varying degradations. For this, we follow closely the work of AirNet [29], and prepare a new test set from BSD68 [41] by applying Gaussian noise of varying levels $\sigma = [0, 15, 25, 50]$ at different spatial locations of the images. Results in Table 8 show that our PromptIR framework is more effective in restoring these images than AirNet [29], providing 0.23 dB improvement.

**Training model with different combinations of degradation.** In Table 1, we report the results of training an all-in-one model on combined datasets from all three restoration tasks. Here, we evaluate the impact on PromptIR performance by different combinations of degradation types (tasks). Table 9) shows that with an increasing number of degradation types, it becomes increasingly difficult for the network to restore images, thereby leading to a performance drop.

Specifically, the presence of hazy images in the combined dataset seems to negatively affect the model. Interestingly, a model trained on the combination of rainy and noisy images achieves good performance, indicating a positive correlation between the deraining and denoising tasks. Such phenomenon is also observed in the AirNet work [29].

**Impact of Prompt Block.** To effectively evaluate the benefit of the proposed framework, we evaluate it against the Restormer model [68]. Restormer is scaled up to match the parameter count of PromptIR. We perform this evaluation under the All-in-one setting. As shown in Table 10, the PromptIR framework provides an average improvement of 0.38 dB over the Restormer model [68].

**Number of prompt components.** We conduct an abalation study to understand the effect of the number of prompt components $P_c$ employed on the performance of the framework. As shown in Table 11, we find that utilizing more than 5 components yields marginal improvements while incurring additional computational overhead. As a result, we choose to utilize five components

Table 10: Comparison of Restormer [68] against the proposed PromptIR framework.

| Method | Dehazing on SOTS [31] | Deraining on Rain100L [16] | Denoising on BSD68 dataset [41]) | | | Average |
|---|---|---|---|---|---|---|
| | | | $\sigma = 15$ | $\sigma = 25$ | $\sigma = 50$ | |
| Restormer[68] | 29.92/0.9703 | 35.56/0.9691 | 33.86/0.9327 | 31.20/0.8878 | 27.90/0.7944 | 31.68/0.910 |
| PromptIR (Ours) | **30.58/0.974** | **36.37/0.972** | **33.98/0.933** | **31.31/0.888** | **28.06/0.799** | **32.06/0.913** |

Table 11: Evaluation on the number of Prompt Components on Rain100L[16]

| Number of Prompt Components | 3 | 5 | 6 | 7 |
|---|---|---|---|---|
| PSNR | 36.91 | 37.04 | 37.06 | 37.07 |

## 5 Conclusion

Existing image restoration models based on deep neural networks can work for specific degradation types and do not generalize well to other degradations. However, practical settings demand the ability to handle multiple degradation types with a single unified model without resorting to degradation-specific models that lack generalization and require apriori knowledge of specific degradation in the input. To this end, our work proposed a drop-in prompt block that can interact with the input features to dynamically adjust the representations such that the restoration process is adapted for the relevant degradation. We demonstrated the utility of prompt block for all-in-one image restoration by integrating it within a SoTA restoration model that leads to significant improvements on image denoising, deraining, and dehazing tasks. In the future, we will extend the model for a broader set of corruptions toward the goal of universal models for better generalization in image restoration tasks.

## Acknowledgement

The computational resources were provided by the National Academic Infrastructure for Super-computing in Sweden (NAISS), partially funded by the Swedish Research Council through grant agreement no. 2022-06725, and by Berzelius resource, provided by the Knut and Alice Wallenberg Foundation at the National Supercomputer Centre.

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

# Appendix

## A    Additional Ablation Studies

We conduct further ablation studies to illustrate the effectiveness of various design choices of the PromptIR framework. We examine various key design choices like the usage of prompt tokens and plugging in prompt blocks only on the decoder branch of the network.

### A.1    Contrastive learning-based Degradation Encoder embedding v/s Prompt Tokens

To strengthen the design rationale for incorporating prompts instead of following recent methods [29] that use embeddings learned through contrastive training, we replace the generated prompt from our PGM module with embeddings extracted from the Contrastive- learning based Degradation Encoder of the AirNet [29] model. We observed that the use of contrastive embeddings resulted in significantly weaker performance compared to prompt tokens. Moreover, achieving good performance with contrastive embeddings requires a custom-designed restoration network, whereas our Prompt Blocks can be seamlessly integrated as plug-and-play modules into any restoration network.

Table A.1: Comparisons under all-in-one setting: between the usage of degradation embedding extracted from the Contrastive-learning Based Degradation Encoder (CBDE) of the Airnet [29] Model and the usage of prompt tokens in the PromptIR framework.

| Method | Dehazing on SOTS [31] | Deraining on Rain100L [16] | Denoising on BSD68 dataset [41]) | | | Average |
| | | | $\sigma = 15$ | $\sigma = 25$ | $\sigma = 50$ | |
|---|---|---|---|---|---|---|
| CBDE+PromptIR | 23.92/0.881 | 32.03/0.0.972 | 32.96/0.910 | 30.36/0.860 | 26.93/0.732 | 29.24/0.875 |
| PromptIR (Ours) | **30.58/0.974** | **36.37/0.972** | **33.98/0.933** | **31.31/0.888** | **28.06/0.799** | **32.06/0.913** |

### A.2    Prompt Blocks on both Encoder branch and Decoder branch

We study the importance of decoder-only prompting by evaluating the usage of prompt blocks on both the encoder and decoder branches. We show that it is important the prompt block is only used on the decoder side.

Table A.2: Comparisons under the all-in-one setting: between the usage of the Prompt-block on both the encoder branch and encoder branch with using the prompt block only on the decoder branch.

| Method | Dehazing on SOTS [31] | Deraining on Rain100L [16] | Denoising on BSD68 dataset [41]) | | | Average |
| | | | $\sigma = 15$ | $\sigma = 25$ | $\sigma = 50$ | |
|---|---|---|---|---|---|---|
| Enc+Dec+PromptIR | 28.52/0.927 | 35.43/0.965 | 33.59/0.927 | 30.85/0.878 | 27.35/0.732 | 31.14/0.885 |
| PromptIR (Ours) | **30.58/0.974** | **36.37/0.972** | **33.98/0.933** | **31.31/0.888** | **28.06/0.799** | **32.06/0.913** |

## B    Transformer Block in PromptIR Framework

As mentioned in section 3.1.2 of the main manuscript, we present the block diagramB.1 of the transformer block and further, elaborate on the details of the transformer block employed in the PromptIR framework. The transformer block follows the design and hyper-parameters outlined in [68]

To begin, the input features $\mathbf{X} \in \mathbb{R}^{H_l \times W_l \times C_l}$ are passed through the MDTA module. In this module, the features are initially normalized using Layer normalization. Subsequently, a combination of $1 \times 1$ convolutions followed by $3 \times 3$ depth-wise convolutions are applied to project the features into Query ($\mathbf{Q}$), Key ($\mathbf{K}$), and Value ($\mathbf{V}$) tensors. An essential characteristic of the MDTA module is its computation of attention across the channel dimensions, rather than the spatial dimensions. This effectively reduces the computational overhead. To achieve this channel-wise attention calculation, the $Q$ and $K$ projections are reshaped from $H_l \times W_l \times C_l$ to $H_l W_l \times C_l$ and $C_l \times H_l W_l$ respectively,

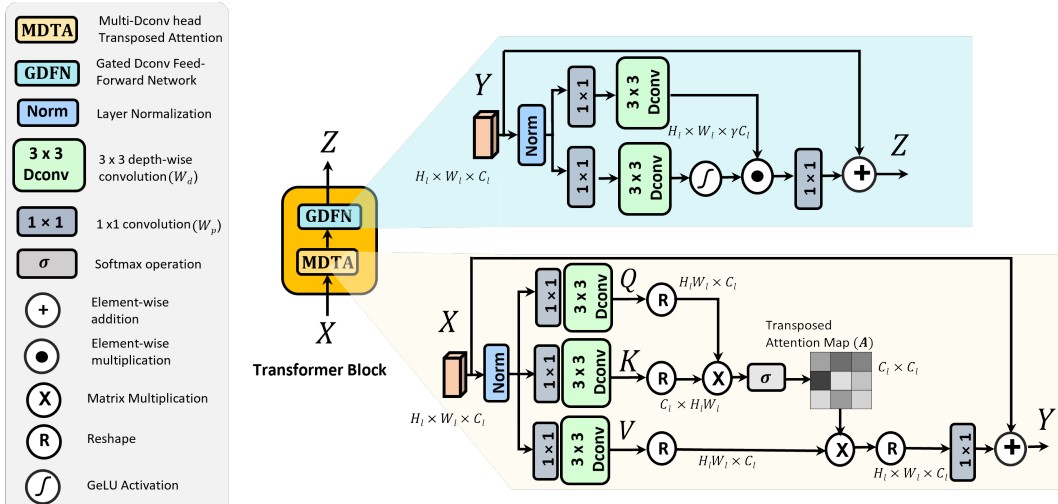

Figure B.1: Overview of the Transformer block used in the PromptIR framework. The Transformer block is composed of two sub modules,the Multi Dconv head transposed attention module(MDTA) and the Gated Dconv feed-forward network(GDFN).

before calculating dot-product, hence the resulting transposed attention map with the shape of $C_l \times C_l$. Bias-free convolutions are utilized within this submodule. Furthermore, attention is computed in a multi-head manner in parallel.

After MDTA Module the features are processed through the GDFN module. In the GDFN module, the input features are expanded by a factor $\gamma$ using $1 \times 1$ convolution and they are then passed through $3 \times 3$ convolutions. These operations are performed through two parallel paths and the output of one of the paths is activated using GeLU non-linearity. This activated feature map is then combined with the output of the other path using element-wise product.

# C    Qualitative results:

We present more qualitative results from single-task models to further elucidate the effectiveness of prompt-block even when under the single-task setting.

## C.1    Dehazing

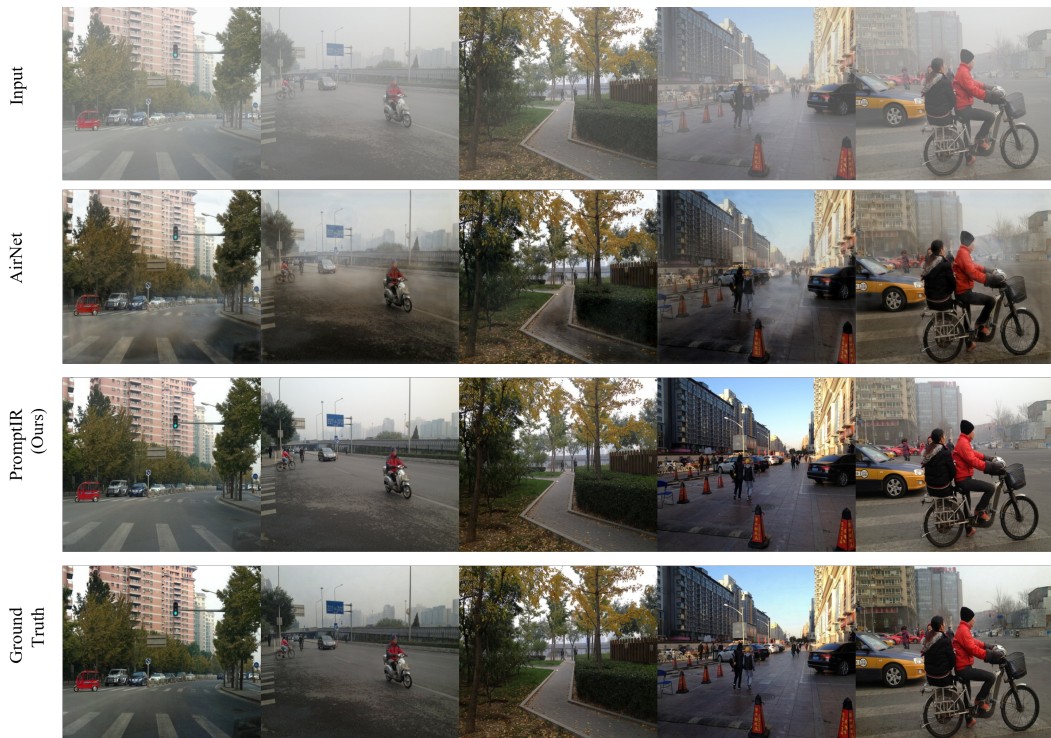

Figure C.1: **Image deraining comparisons** under single task setting on images from the SOTS dataset [31]. Our method effectively removes haze to produce visually better images.

## C.2   Deraining

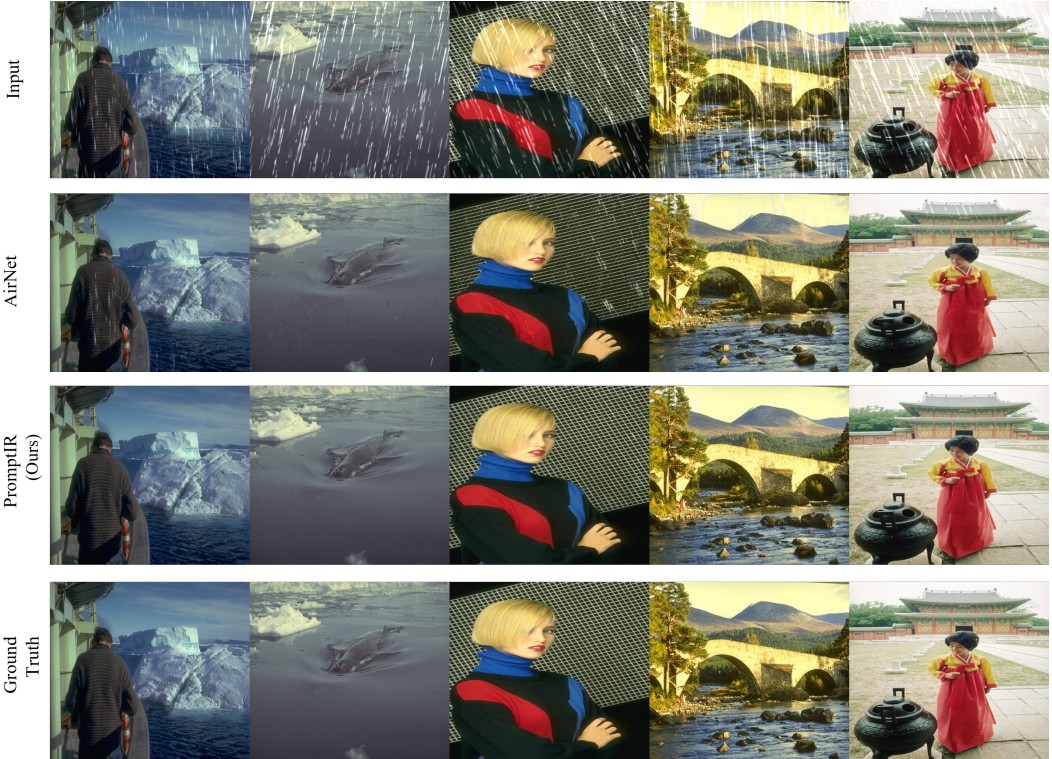

Figure C.2: **Image deraining comparisons** under single task setting on images from the Rain100L dataset [16]. Our method effectively removes rain streaks to generate rain-free images.

## C.3 Denoising

| Input Image | Noisy Image | Ground truth | AirNet | PromptIR(Ours) |
|---|---|---|---|---|

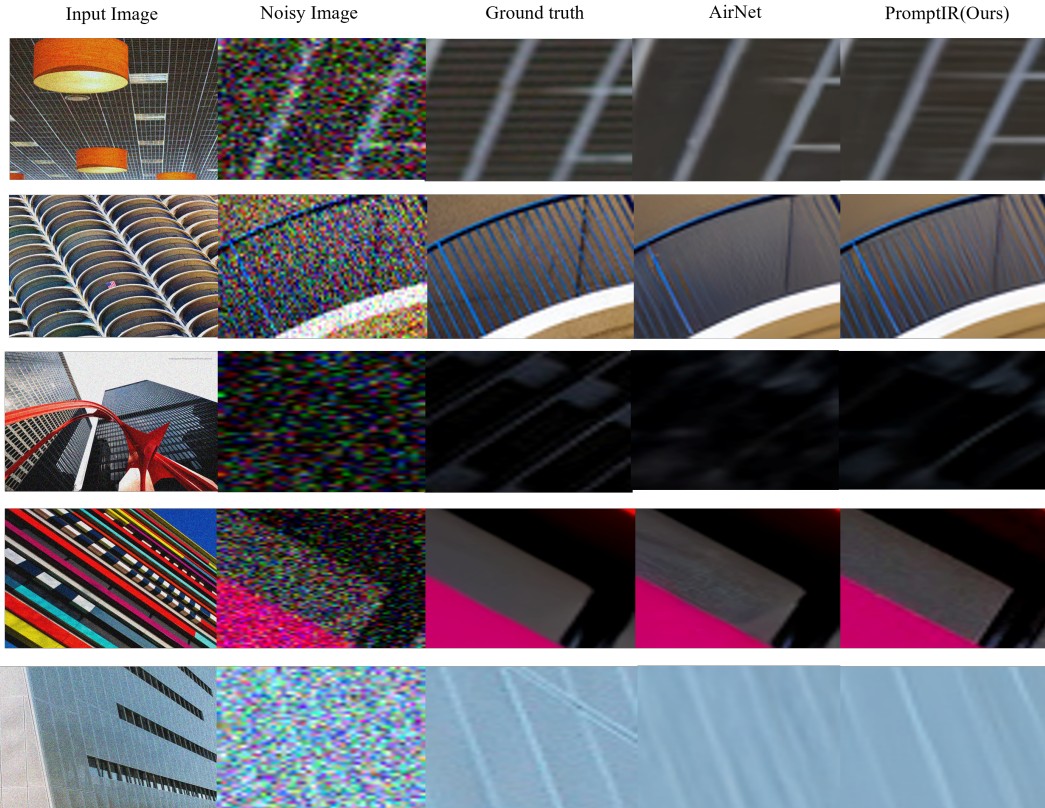

Figure C.3: **Image deraining comparisons** under single task setting on images from the URBAN100 dataset [22] with $\sigma = 50$. Our method produces visually better images as compared to previous methods. We show selected patches from the images.

