# OpenReview forum: "PromptIR: Prompting for All-in-One Image Restoration"
_NeurIPS.cc/2023/Conference — NeurIPS 2023 poster_

### Official Review · Reviewer_4MQM · 2023-06-30

**Soundness:** 3 good
**Presentation:** 3 good
**Contribution:** 3 good
**Rating:** 5
**Confidence:** 4

**Summary:**

The paper proposes PromptIR, a prompt-based learning approach for all-in-one image restoration. The method utilizes prompts to encode degradation-specific information and dynamically guide the restoration network. It achieves state-of-the-art results on various image restoration tasks.

**Strengths:**

(1) The proposed method offers a generic and efficient plugin module with a few lightweight prompts that can be used to restore images of various types and levels of degradation with no prior information of corruption.

(2) The method achieves state-of-the-art results in image denoising, deblurring, and dehazing.

(3) The paper is well-written and easy to follow.


**Weaknesses:**

(1) The paper lacks discussions on the limitations of the proposed method, such as the sensitivity of the results to the number of prompts used and how the method performs on combined types of degradation, such as noise + rain.

(2) The paper lacks the visualization of the learned prompt and an explanation of how and why it could work.

(3) The paper lacks comparisons on complexity, including runtime and parameter number.


**Questions:**

(1) Could you please provide more details on the sensitivity of the results to the number of prompts used and how to choose the optimal number of prompts?

(2) How does the Restorformer perform in the setting of Table 1?

(3) How about the perceptual metric comparison between this method and the baselines?


**Limitations:**

See the weakness and question part.

---

> ### Author Rebuttal · Authors · 2023-08-10
>
> `R5.1`: *Ablations on number of prompt blocks and multi-degraded images*:
>
> The number of prompt components is chosen empirically. Please refer to response 2.4 (Table R3). And for a detailed response on images corrupted with multiple degradations, please see R1.3.
>
> `R5.2`: *Learned prompts*:
>
> We perform an ablation experiment by replacing prompt blocks with restormer blocks, making the params/FLOPs similar to our PromptIR framework. The new baseline network yields PSNR of 36.82 dB, whereas our PromptIR provides 37.04 dB.
>
> `R5.3`: *Computational comparisons*:
>
> The parameters, Flops and runtimes are reported in Table R1.
>
> `R5.4`: *Number of prompts*:
>
> We empirically choose number of prompt components. Please refer to R2.4 for more details
>
> `R5.5`: *Results of Restormer*:
>
> When averaged across all three tasks, our method provides 0.38 dB performance gain over Restormer, as shown in Table R5.
>
> | Method    | Dehazing(SOTS) | Deraining(Rain100L)        |     |  Denoising(BSD68)   |      | Average       |
> |------------|-------------------|------------------|----------------------------|------|----|----------------|
> |                 |                           |                         | $\sigma = 15$          | $\sigma = 25$ | $\sigma = 50$ |               |
> | Restormer | 29.92/0.9703      | 35.56/0.9691      | 33.86/0.9327           | 31.20/0.8878 | 27.90/0.7944  | 31.68/0.910   |
> | PromptIR  | **30.58**/**0.974** | **36.37**/**0.972** | **33.98**/**0.933** | **31.31**/**0.888** | **28.06**/**0.799** | **32.06**/**0.913** |
>
>
> **Table R5**: Add Restomer results in Table 1 of the main paper
>
> `R5.6`: *LPIPs metric*:
>
> Our PromptIR provides better image fidelity scores using LPIPs than AirNet, as shown in Table R6.
>
> | Method    | Denoising on BSD68 | Deraining on RAIN100L | Dehazing on SOTS |
> |-----------|---------------------|-----------------------|------------------|
> | AirNet    | 0.1961              | 0.0525                | 0.0431           |
> | PromptIR  | 0.1958              | 0.0408                | 0.0168           |
>
> **Table R6**: LPIPs scores

---

### Official Review · Reviewer_vXsu · 2023-07-02

**Soundness:** 3 good
**Presentation:** 3 good
**Contribution:** 3 good
**Rating:** 5
**Confidence:** 5

**Summary:**

This paper proposes an all-in-one image restoration method that can restore images from various types and levels of degradation. Specifically, it utilizes prompts to encode degradation-specific information, which is used to dynamically guide the restoration network. Experimental evaluations on image denoising, deraining, and dehazing are conducted to demonstrate the effectiveness of the proposed method.

**Strengths:**

Prompts are used to encode degradation-specific information for all-in-one image restoration.

Experimental evaluations and ablation studies are conducted to demonstrate the superior performance of the proposed method.

**Weaknesses:**

There are several unclear statements listed as follows.

**Questions:**

1. What do the degradation embeddings used in Figure 2 refer to?
2. What does the feature F_p in Figure 3 refer to?
3. The key component of the proposed approach is the prompt block, which takes the prompt-components P_c and input features F_l as input. However, few information about P_c is provided. Please explain in detail how do the authors obtain the prompt-components P_c.
4. Do the authors retrain all the competed methods using the same training dataset as the proposed method?
5. For the single task setting, more recent and representative methods should be compared to demonstrate the effectiveness of the proposed method. For example,
1) for image dehazing,
      a) Song et al., Vision Transformers for Single Image Dehazing, IEEE TIP, 2023.
      b) Tu et al., MAXIM: Multi-Axis MLP for Image Processing, CVPR 2022.
2) for image deraining,
      c) Chen et al., Pre-Trained Image Processing Transformer, CVPR 2021.
6. To show the impact of PGM, it would be better to compare with a baseline that replaces all the prompt blocks in Figure 3 with the transformer blocks in Table 5.

**Limitations:**

This paper does not claim the limitations and potential negative societal impact of their work. It would be better to discuss about the limitations in real applications and potential negative societal impact of their work.

---

> ### Author Rebuttal · Authors · 2023-08-10
>
> `R4.1`: *Degradation embeddings*:
>
> In Figure 2, we plot the prompt P ,generated through the Prompt Generation Module within the Prompt block. Here, the term ”degradation embedding” is used to refer to the generated prompt P, as it provides
> additional contextual information regarding the type of degradation observed within a corrupted image.
>
> `R4.2`: *Typo*:
>
> Sorry for the typo. It should be $F_l$. We will fix it.
>
> `R4.3`: How prompt-components $P_c$ are obtained?
>
> Prompt components are learnable parameters that are initialized with random weights (for instance using nn.Parameter(*size) in Pytorch).
> Subsequently, these prompts undergo optimization throughout the training process to acquire specialized contextual information related to degradations. At inference time, these prompts guide the model in restoring degraded images. As recommended, we will provide a more detailed explanation of prompt components in the revised paper.
>
> `R4.4`: *Training setting*:
>
> We follow the experimental protocols and datasets defined in the AirNet paper, and extended their tables with our results to avoid retraining any previous approach.
>
> `R4.5`: *Further comparisons*:
>
> We thank reviewer for the references. We will cite the work of Song et al., while the others are already credited. Including these referenced methods would require retraining them. Our PromptIR approach is based on the Restormer baseline. Given that Restormer has demonstrated superior outcomes compared to MAXIM and IPT, we are confident that our approach will likely exhibit a similar advantageous trend.
>
> `R4.6`: *Baseline*:
>
> As per reviewer suggestion, we perform this ablation experiment by replacing prompt blocks with restormer blocks, making the params/FLOPs similar to our PromptIR framework. The new baseline network yields PSNR of 36.82 dB, whereas our PromptIR provides 37.04 dB.

---

> > ### Comment · Reviewer_vXsu · 2023-08-19
> > **My concerns are addressed.**
> >
> > Thanks for the response, which solves my questions. I also read other reviewers’ comments and the corresponding responses. Overall, I am satisfied with the response. I would like to keep my score.

---

### Official Review · Reviewer_UJGQ · 2023-07-05

**Soundness:** 3 good
**Presentation:** 2 fair
**Contribution:** 2 fair
**Rating:** 5
**Confidence:** 5

**Summary:**

In this paper, authors study to restore images with unknown degradation types with a unified framework. To achieve it, a prompt-based block is proposed which generates suitable prompt first and then interact them with original features. Experiments show that combining the proposed block with restormer network can achieve new SOTA performance.

**Strengths:**

1. Prompt learning is recent a hot method in both language and vision domains. Different from previous works focus on high-level understanding tasks, this paper tempts to explore its application on low-level image restoration task.
2. The writing is good and easy to understand.

**Weaknesses:**

1. Typos or confused explanations:
     a. In Eq. 2, subscript of w should be the same with P in $\sum_{c=1}^{N} w_{I}P_{c}$? And the subscript of w in the second formula should be removed?
     b. Line 208, "Table 1 Similarly on the image Deraining task, ".
     c. Line 204, the second best approach is not DL in Table 1, is MRPNet.
     d. Line 231-232, it mentions that using only one prompt block in the latent space (36.76 dB) degrades the performance. However, the baseline is 36.74 dB, which lower than 36.76 dB.
2. Lack some comparisons. For example, the performance restormer is needed in Table 1. And other all-in-one weather image removing methods[1][2] are still very related to this work and needed to be compared. In addition, all results are in the synthesized images, how about its general performance on real world images (e.g., real rain images).
3. Ablation study can be further improved. For example, the number of prompt is not studied. And the additional computational cost and model parameters should be discussed. Is the improvement gains come from the additional parameters?
[1] Transweather: Transformer-based restoration of images degraded by adverse weather conditions;
[2] Learning Multiple Adverse Weather Removal via Two-Stage Knowledge Learning and Multi-Contrastive Regularization: Toward a Unified Model;

**Questions:**

I think applying prompt technology into low-level image restoration task is very interesting, but this work seems suffer from many issues. Some of them are mentioned in the weaknesses. Still, there are some other questions as followed:

1. As mentioned in the paper, the proposed prompt block can be used in various existing networks. However, no experiments are conducted. And the ablation studies in table 6 and table 2 (supplementary) show that different networks may require various block configurations.
2. Why not use cross attention to interact learnable prompt with features? In [1], it seems work well. Thus, I suggest to compare this two ways in detail (the learnable parameters in [1] can be also regarded as prompt).
[1] Transweather: Transformer-based restoration of images degraded by adverse weather conditions;



**Limitations:**

Authors have not discussed the limitations. In my opinions, one of the potential limitations is the unproved generalization ability in terms of networks (e.g., utilization on AirNet) and data (e.g., real world images).

---

> ### Author Rebuttal · Authors · 2023-08-10
>
> `R3.1`: *Typos*:
>
> We thank the reviewer for pointing out these issues. We will improve clarity and fix typos in the revised manuscript.
>
> `R3.2`: *Additional Comparisons*:
>
> As suggested, we include Restormer and Transweather in comparisons. Please refer to Table R1. Our PromptIR provides favorable results.
>
> `R3.3`: *Ablation Study*:
>
> Please refer to Table R3 for ablation on number of blocks, and Table R1 for the computational comparisons. These will be included in the revised manuscript.
> As suggested, we perform an ablation experiment in which we adjust the baseline Restormer's capacity such that the params/FLOPs become similar to our PromptIR framework. This ensures that the performance gains are obtained due to the novel architecture design rather than the extra parameters. This baseline network yields PSNR of 36.82 dB, whereas our PromptIR provides 37.04 dB.
>
> `R3.4`: *Further Questions*:
>
> Majority of recent image restoration works (Restormer, HINet, NAFNet, MAXIM, MPRNet) employ a hierarchical architecture. In our PromptIR, we also use UNet design and demonstrate how to simply plug PGM and PIM into the main network. This can easily be extended to other networks as well.
> Note that our prompt block is already a kind of cross-attention because the interaction between features and prompts is taking place with a transformer block.

---

> > ### Comment · Reviewer_UJGQ · 2023-08-14
> >
> > Thanks for the detailed clarification from the authors. Considering other reviewers' comments, I'd like to increase my final decision as borderline accept.

---

### Official Review · Reviewer_3Jtv · 2023-07-06

**Soundness:** 2 fair
**Presentation:** 2 fair
**Contribution:** 2 fair
**Rating:** 4
**Confidence:** 5

**Summary:**

This paper restores images from various types and levels of degradation. In detail, degradation information is encoded as prompts and used to guidance the restoration. Based on the classical UNet architecture for restoration, there are two added blocks: prompt generation module and prompt interaction module. With a single network, it achieves state-of-the-art performance on image denoising, deraining and dehazing.

**Strengths:**

1, A prompting-based single-model blind restoration framework.

2, A plug-in module with prompt generation and prompt interaction, which guide the backbone model to remove corruptions effectively.

3, State-of-the-art performance on multiple tasks with one single model

**Weaknesses:**

1, The ablation studies cannot prove this basic concern of the main contribution and intuition in this paper. Is the prompt (5 learned tensors as a dictionary) really working? What have the prompts learned? Do we really need the prompts? Since there are only 6 different types of degradations, I guess $softmax(Conv_{1\times 1}(GAP(F_1)))$ learns a classification model to decide which degradation the input image has. If we directly inject this information to the UNet, it should work as well. This paper still follows the pipeline of "degradation estimation + degradation-aware restoration".

2, Missing comparison on model size, runtime, FLOPs, etc. According to the model file sizes, I suspect there exists unfair comparison. The model size of PromptIR (file size: 388MB) is about 4 times of Restormer (file size: 99.9MB) and 11 times of Airnet (file size: 35.6MB). This means that the model might use 100MB for each sub-task and the rest 88MB for estimating the degradation type.

3, For the single-task setting, for example, image denoising with sigma=50, what's the meaning of using prompt generation and interaction?

4, Missing ablation studies on prompt design, such as number of prompts.

5, Why adding prompt blocks at level 4 is nearly of no use (36.76 v.s. 36.74dB)? Note that it has more parameters in the prompt generation module and prompt interaction module. Besides, how about using level 2 only?

6, In table 7 and 8, why PromptIR is significantly better than Airnet on unseen noise level, but is only slightly better on spatially variant degradation?



**Questions:**

See weakness

**Limitations:**

See weakness

---

> ### Author Rebuttal · Authors · 2023-08-09
>
> `R2.1`: *Ablation study (directly incorporating softmax outputs):*
>
> To show the benefit and contribution coming from only prompts in our design, we perform an ablation experiment in which we adjust the baseline Restormer's capacity such that the params/FLOPs become similar to our PromptIR framework. This ensures that the performance gains are obtained due to the novel architecture design (prompt components) rather than the extra parameters. This baseline network yields PSNR of 36.82 dB, whereas our PromptIR provides 37.04 dB. This shows the contribution of prompts towards generic image restoration task.
>
> Secondly, PromptIR does not estimate degradations **explicitly**. Instead, it implicitly calculates these degradations, which subsequently help the main network in restoring images.
>  As suggested, we remove prompt components and instead directly incorporate the information after the softmax into the main architecture. This newly trained network achieves 36.79 dB PSNR, lower by 0.25 dB compared to PromptIR's performance.
>
> `R2.2`: *Computational Comparison*:
>
> Our PromptIR is an all-in-one network, and does not hold separate parameters for separate tasks, i.e., incorporating additional tasks into the pipeline would not lead to increase in parameters. The public checkpoints
> for Restormer and AirNet do not include the optimizer state dictionary,
> resulting in smaller file sizes. Similarly, if memory conservation is a priority, one can opt to exclude the optimizer dictionary from PromptIR. For computational comparisons, please refer to Table R1.
>
> `R2.3`: *Prompting for single task*:
>
> We followed AirNet to further include single-task experiments in the paper. In our tests, we noticed that the use of prompts also contributes
> to the improvement of single-task restoration. These prompts provide
> complementary information to the main network, enabling it to better
> distinguish various elements such as rain patterns and streaks of different
> orientation and magnitude in the context of deraining.
>
> `R2.4`: *Ablation on prompt components*:
>
> As suggested, we conduct an ablation study on the number of prompt
> components, observing that employing more than 5 components yields
> marginal improvements while incurring additional computational overhead. As a result, we choose to utilize five components, a decision supported by our findings. We will include this ablation Table R3 in the
> revised paper.
>
> | # of prompt components | 3     | 5     | 6     | 7     |
> |-----------------------|-------|-------|-------|-------|
> | PSNR                  | 36.91 | 37.04 | 37.06 | 37.07 |
>
> **Table R3**:  Ablation study on the number of prompt components.
>
> `R2.5`: *Using prompts on only level 2 or level 4*:
>
> When employing the prompt block solely at level 2, the PSNR drops from 37.04 dB to 36.82 dB. We tried several combinations for placing prompt block, and empirically found that placing prompt blocks between every consecutive level of the decoder performs the best.
> Although the improvement only with level 4 is not much, keeping the block at this level is important since it feeds to later restoration stages and allows overall better performance working in tandem with other levels.
>
> `R2.6`: *Spatially variant degradation*:
>
> To truly test the generalization capability of our PromptIR, we reported results on unseen noise level of 100 (Table 7), where our method shows a significant gain of ~7 dB over AirNet. Whereas in Table 8, the noise levels used in the spatially variant degradation experiment are seen by both AirNet and our PromptIR during their training phases. As a result, the performance gap between the two methods was narrower. Here, we perform a new spatially variant degradation experiment by including both seen and unseen noise levels $\sigma$=[0, 50, 75, 100]. Our method yields a significant PSNR gain of 5 dB (i.e., 22.34 dB ours vs 17.35 dB Airnet).
>
> | Method   |  PSNR  | SSIM   |
> |----------|--------|--------|
> | Airnet   | 17.35  | 0.5420 |
> | PromptIR | 22.34  | 0.6137 |
>
> **Table R4**: Additional ablation on spatially-variant degradation.

---

> > ### Comment · Reviewer_3Jtv · 2023-08-20
> >
> > The key concern is my first point, which is not addressed in your reply. The authors didn’t answer “what have the prompts learned?” or give any meaningful assumptions. After all, it is counterintuitive to believe that 5 learned fixed tensors can boost the performance for 0.25db.
> >
> > Therefore, I keep my score and tend to reject this paper as my concerns are not resolved and the used prompt mechanism is counterintuitive and unexplainable in this paper.
> >
> > 1, what is the setting of retraining a Restormer with similar capacity? Did you train it on single task or multiple task? If on single task, it is hard to believe that one model trained for a single task is worse. If on multiple tasks, did you use any conditioning method to help the model deal with different tasks?
> >
> > 2, what does “implicitly calculate these degradation” mean? What do the prompts learn? (Degradation representation?) How did you remove the prompt components? (By randomly initialize it and freeze it?)
> >
> > 3, For single-task testing, prompts are used for deraining. Did you also use it for denoising 50? (My original question. Gaussian denoising uses an uniform degradation. The degradation information for all images and all areas should be same.

---

> > > ### Author Response · Authors · 2023-08-21
> > >
> > > `A2.1`: *What have prompts learned?*
> > >
> > > Figure 2 of the paper shows a t-SNE plot of the degradation embeddings
> > > of the proposed PromptIR. This plot highlights that the output prompts generated by the PGM effectively encode discriminative information regarding the
> > > corruptions present in the input images. Additionally, these generated prompts
> > > (learned in the input space) provide degradation-aware context to the PIM module, which dynamically adapts the input features for an improved restoration
> > > procedure.
> > >
> > > In light of the reviewer’s question (’what have the prompts learned’), in addition to the previously provided ablation experiment, we have made our best efforts to visualize the individual prompt components. However, the complexity of the visualization maps makes their interpretation challenging, preventing
> > > definitive conclusions. Note that prompts have been widely used for several
> > > other problems, however, we find *no* examples from previous literature directly visualizing individual prompt components to illustrate their encoded information [A-F]. This shows that it is not trivial to directly visualize what have
> > > the prompts learned, however, Fig. 2 visualization shows their discriminative
> > > behavior and our results show their effectiveness.
> > >
> > > [A] Learning to Prompt for Vision-Language Models
> > >
> > > [B] PIVOT: Prompting for Video Continual Learning
> > >
> > > [C] DualPrompt: Complementary Prompting for Rehearsal-free Continual Learning
> > >
> > > [D] Learning to Prompt for Continual Learning
> > >
> > > [E] MaPLe: Multi-modal Prompt Learning
> > >
> > > [F] Conditional Prompt Learning for Vision-Language Models
> > >
> > >
> > > `A2.2`: *Experimental settings for retraining Restormer.*
> > >
> > > For this ablation experiment, we retrained Restormer for a single-task setting. We increase the baseline Restormer’s capacity by adding more Transformer blocks, thereby bringing the params/FLOPs to a comparable level as our
> > > PromptIR framework. Quantitative results show that this baseline yields lower
> > > PSNR in comparison to our PromptIR. It’s important to note that, throughout
> > > this ablation, both the Restormer baseline and the proposed PromptIR were
> > > subjected to exactly identical experimental configurations. All our codes and
> > > trained models will be released to ensure reproducibility.
> > >
> > > `A2.3 (a)`: *Implicit degradation estimations means?*
> > >
> > > The statement “Our PromptIR implicitly estimates degradation” can be
> > > clarified as follows: We do not provide any explicit ground-truth supervision
> > > regarding the degradation types and levels to the PromptIR framework. Instead,
> > > the framework acquires this information implicitly from the data and encodes
> > > it in learned prompts.
> > >
> > > `A2.3 (b)`: *How were prompt components removed?*
> > >
> > > First, we remove the prompt components branch entirely from the PGM
> > > module. Next, the information after the softmax is passed to the PIM that
> > > modulates the input features via element-wise multiplication before processing
> > > these features further through the Transformer blocks and convolution layers
> > > present in the PIM module.
> > >
> > > `A2.4`: *Are prompts used for single-task denoising as well?*
> > >
> > > Yes, the prompt blocks are also used in denoising (σ=50). In restoration
> > > tasks, where the degradation impacts the image content in a non-homogenous
> > > manner (e.g. dehazing and deraining), the proposed PromptIR modules exhibit
> > > their true potential, thus contributing significantly in achieving substantial performance gains. Whereas, as pointed out by the reviewer, the performance gains
> > > are comparatively less pronounced in the denoising task, where the degradation
> > > affects all areas of the image uniformly.

---

> > > > ### Comment · Reviewer_3Jtv · 2023-08-21
> > > >
> > > > Thanks for your reply. Overall, what the prompts have learned is still unconvincing. I feel like it is more similar to dynamic convolution, rather than prompts in NLP. Here are some following questions.
> > > >
> > > > 1, What are visualised in Fig. 2? 5 learned prompts or degradation embeddings after plug-in prompt blocks? I guess it is the later one. In this case, I think the better clustered embeddings can prove that the plug-in prompt blocks (i.e., the "feature classification module") works well and can classify the inputting $F_I$ as Hazy, Rainy and Noisy. I don't think it's the contribution of the 5 learned prompts.
> > > >
> > > > 2, As for the cited papers [A-F], I guess they don't have such an interpretation problem, as the learned prompt is a special text embedding with specific meaning. Here, the learned prompts may not encode the context information. Most discriminative information comes from another input $F_I$.
> > > >
> > > > 3, If Restormer is trained with similar capacity and achieves worse performance, I guess it is because the "plug-in prompt blocks" works like an attention mechanism (learn to classify the feature and then multiply the score with the original features. See SENet.). Attributing the performance gain to the learned prompts might be misleading.
> > > >
> > > > 4, For the PGM module, can we explain it as a dynamic convolution module? The learned prompts are like learned convolution kernels, which are weighted by $F_I$.

---

> > > > > ### Author Response · Authors · 2023-08-22
> > > > >
> > > > > We would like to thank the reviewer for their response.
> > > > >
> > > > > `B2.1`: *What is visualized in Fig.2?*
> > > > >
> > > > > The embeddings visualized in Fig. 2 are the generated prompts P , that are
> > > > > the outputs of the PGM module. Since the PGM module is directly dependent
> > > > > on the 5 learned prompt-components, the better quality of the embeddings can
> > > > > be attributed to the usage of these learned parameters. Please refer further to
> > > > > response B2.3 below for quantitative evidence.
> > > > >
> > > > > `B2.2`: *Prompt learning in computer vision.*
> > > > >
> > > > > Prompt learning-based paradigms are not only used in NLP but also explored
> > > > > in vision problems. Techniques rooted in prompt learning, wherein adaptable
> > > > > parameters known as soft-prompts are employed to enrich contextual information, have demonstrated their efficacy across diverse computer vision tasks. For
> > > > > example, in image classification [G], video continual learning [H], multi-modal
> > > > > learning [I], continual learning for image classification [J,K,L]. Again, we will
> > > > > like to clarify that MaPLe and VPT both use prompts with image encoders
> > > > > and have no direct evidence to explain exactly what information is encoded in
> > > > > these visual prompts. We believe this is an open research question to interpret and explain the information encoded in prompts.
> > > > >
> > > > > [G] Visual Prompt Tuning.
> > > > >
> > > > > [H] PIVOT: Prompting for Video Continual Learning.
> > > > >
> > > > > [I] MaPLe:Multi-modal Prompt Learning.
> > > > >
> > > > > [J] Learning to Prompt for Continual Learning.
> > > > >
> > > > > [K] DualPrompt: Complementary Prompting for Rehearsal-free Continual learning.
> > > > >
> > > > > [L] CODA-Prompt: Continual Decomposed Attention-based Prompting for Rehearsal-Free Continual Learning.
> > > > >
> > > > > `B2.3`: *Comparison with SENet.*
> > > > >
> > > > > In the experiment of response R2.1, when we remove the prompt components, the PGM module already reduces to nearly the SENet module. Now the PGM (without prompt components) first computes an attention vector (squeeze
> > > > > operation) from the incoming features maps, which is later used to excite the
> > > > > input features at the onset of PIM. The outcome of this experiment provided
> > > > > 0.25 dB lower PSNR compared to the network that incorporates prompt components, thereby indicating that the performance gains are indeed linked to the
> > > > > presence of prompt components.
> > > > >
> > > > > `B2.4`: *Prompt components vs dynamic convolution.*
> > > > >
> > > > > Dynamic convolution is strictly input dependent, i.e. it is computed from
> > > > > the incoming input features; which is not the case with our prompt-components.
> > > > > Instead, prompt components are independently initialized and later enriched
> > > > > with degradation-aware context during training.

---

> ### Author Response · Authors · 2023-08-18
> **We thank the reviewer again for the valuable feedback and happy to address any remaining concerns.**
>
> We extend our sincere gratitude to the reviewer for their valuable time and insightful feedback. We value your constructive feedback and hope that our responses have appropriately addressed all the concerns.
>
> We really appreciate the valuable time to respond to our feedback based on the reviewer's comments. Further, we are happy to address any remaining concerns.

---

### Official Review · Reviewer_Jp7P · 2023-07-09

**Soundness:** 3 good
**Presentation:** 3 good
**Contribution:** 3 good
**Rating:** 5
**Confidence:** 5

**Summary:**

The paper proposes prompt based image restoration method that can handle different degradations. the proposed method uses prompts to encode degradation-specific information, which is then used to dynamically guide the restoration network. Given any instance, prompt generation module (PGM) and a prompt interaction module (PIM) are used generate input-conditioned prompts (via PGM) that are equipped with useful contextual information to guide the estoration network (with PIM) to effectively remove the corruption.

**Strengths:**

- Paper is well organized
- PGM, and PIM modules are used to generate the prompts and recover the image. Proposed method is clearly explained


**Weaknesses:**

- Can the proposed method method handle degradations like snow, night, blur, and turbulence conditions, since the proposed method mentions it is generic image restoration.
- experiments are limited, since the experiments showed in the paper covered only haze, rain, and noise, specifically these the datasets used for comparison are synthetic and very small datasets. it is hard understand the limits of the proposed method from these experiments
- Additionally the experiments performed contains only single degradation at any instance, can the proposed method handle multiple degradations in the single input image
- Can authors performs the experiments showing if the wrong prompt was propagated (instead of the prompt from the PGM) to understand the importance of PGM module.


**Questions:**

Please refer weaknesses

**Limitations:**

Limitations of the proposed method were not discussed

---

> ### Author Rebuttal · Authors · 2023-08-09
>
> `R1.1`:*Other Degradations*:
>
> We opted for deraining, dehazing and denoising to demonstrate the effectiveness of our approach on a diverse range of low-level vision tasks. However, as per the reviewer suggestions, we retrained our PromptIR network on five distinct tasks, including two new tasks i.e, low-light enhancement and deblurring. We followed the recently published work [1] and supplemented their table with our obtained results. Table R1 shows that our method yields favorable performance. (Please note that these results are from our initial attempt in a limited time, which can further be improved with more careful choices of hyperparameters.)
>
> | Method      | Deraining | Dehazing | Denoising | Deblurring | Low-Light | Average PSNR | Param | FLOPs (G) | Runtime (ms) |
> |-------------|-----------|----------|-----------|------------|-----------|--------------|---------|-----------|--------------|
> | DL          | 21.96     | 20.54    | 23.09     | 19.86      | 19.83     | 21.05   | 2.09M    | -         | -            |
> | Transweather| 29.43     | 21.38    | 29.00     | 25.12      | 21.21     | 25.22  | 37.93M  |  4.684     | 14.0         |
> | Airnet      | 32.98     | 21.04    | 30.91     | 24.35      | 18.18     | 25.49  | 8.93M   |  311       | 141.0      |
> | Restormer   | 34.81     | 24.09    | 31.49     | 27.22      | 20.41     | 27.60  | 26.13M  |  155       | 87.9       |
> | IDR         | 35.63     | 25.24    | 31.60     | 27.87      | 21.34     | 28.34  | 15.34M  | -         | -            |
> | PromptIR    | 36.09     | 30.60    | 31.26     | 27.47      | 23.03     | 29.46 | 35.6M   |  173       | 95.5       |
>
> **Table R1**: FLOPs and Runtimes are computed on 256x256 images.
>
>
> `R1.2`: *Experiments*:
>
> In this paper, we adhered to the experimental protocols, datasets, and
> tasks outlined by the previous best method AirNet (CVPR’22). It was
> to ensure fair comparisons as well as avoid retraining all previous works.
> Nonetheless, it is important to highlight that the increase in dataset scale
> will likely provide improved results, which is an expected outcome of deep
> learning techniques. In this response, we have further provided additional
> results as suggested by reviewers which demonstrate the scalability and
> efficacy of PromptIR (Table R1).
>
>
>
> `R1.3`: *Multiple degradations in single image*:
>
> The proposed PromptIR is all-in-one method that can recover images with-
> out prior knowledge of input degradation type. It is important to clarify
> that, as in AirNet, “multiple-degradation” indicates that an individual image contains only a single distinct type of degradation, while the dataset will encompass multiple such degradations. In order to handle multiple
> degradations in the single input image, our model and other competitors
> will need to be trained in a similar setting (i.e., multiple degradations per
> image) which we are currently running, and we can include a comparison
> in the final version. In order to cater reviewer’s suggestion, we perform the
> following experiment without any training on multiple degradations per
> image. We add Gaussian noise to Rain100L images to create rainy+noisy
> dataset. Next, we take the pretrained AirNet and our PromptIR and di-
> rectly evaluate on this dataset. Results in Table R2 demonstrates that
> our algorithm fares better than the AirNet method.
>
>
> | Method   | Rain + σ = 15 | Rain + σ = 25 | Rain + σ = 50 |
> |----------|--------------|--------------|--------------|
> | Airnet   | 18.45        | 17.30        | 14.65        |
> | PromptIR | 18.85        | 17.88        | 15.76        |
>
>
>
>
>
> **Table R2**: Evaluation on single-image corrupted with multiple degradations (Rain + Noise).
>
>
> `R1.4`: *Experiment with wrong prompt inputs*:
>
> In ablation study (Table 5 main paper), we show the PGM contributes
> favourably towards the final performance of our method. As per suggestion,
> we added random noise to perturb the generated prompts at inference time
> and there is a significant drop in PSNR from 37.04 dB to 36.89 dB.
>
>
> [1] Zhang et al., Ingredient-oriented Multi-Degradation Learning for Image Restoration, CVPR 2023

---

### Decision · Program_Chairs · 2023-09-21

**Decision:**

Accept (poster)

**Comment:**

The paper received 1xBorderline Reject and 4xBorderline Accept. The authors provided responses.

After carefully reading the paper, the reviews, as well as the authors' responses and discussions, the ACs agree with the majority of the reviewers that the work has merits and can be accepted for publication.

The authors are asked to further refine their paper for camera ready with the information provided in their responses and by using the received feedback from the reviewers.